# Induction of Brain Insulin Resistance and Alzheimer’s Molecular Changes by Western Diet

**DOI:** 10.3390/ijms23094744

**Published:** 2022-04-25

**Authors:** Anna Mietelska-Porowska, Justyna Domańska, Andrew Want, Angelika Więckowska-Gacek, Dominik Chutorański, Maciej Koperski, Urszula Wojda

**Affiliations:** Laboratory of Preclinical Testing of Higher Standard, Nencki Institute of Experimental Biology Polish Academy of Sciences, 02-093 Warsaw, Poland; a.mietelska@nencki.edu.pl (A.M.-P.); j.domanska@nencki.edu.pl (J.D.); a.want@nencki.edu.pl (A.W.); a.wieckowska@nencki.edu.pl (A.W.-G.); chutoranski.dominik@gmail.com (D.C.); maciek-koperski@wp.pl (M.K.)

**Keywords:** Alzheimer’s disease, Western diet, insulin signaling, brain insulin resistance

## Abstract

The term Western diet (WD) describes the consumption of large amounts of highly processed foods, rich in simple sugars and saturated fats. Long-term WD feeding leads to insulin resistance, postulated as a risk factor for Alzheimer’s disease (AD). AD is the main cause of progressive dementia characterized by the deposition of amyloid-β (Aβ) plaques and neurofibrillary tangles consisting of the hyperphosphorylated tau (p-Tau) protein in the brain, starting from the entorhinal cortex and the hippocampus. In this study, we report that WD-derived impairment in insulin signaling induces tau and Aβ brain pathology in wild-type C57BL/6 mice, and that the entorhinal cortex is more sensitive than the hippocampus to the impairment of brain insulin signaling. In the brain areas developing WD-induced insulin resistance, we observed changes in p-Tau(Thr231) localization in neuronal subcellular compartments, indicating progressive tauopathy, and a decrease in amyloid precursor protein levels correlating with the appearance of Aβ peptides. These results suggest that WD promotes the development of AD and may be considered not only a risk factor, but also a modifiable trigger of AD.

## 1. Introduction

Alzheimer’s disease (AD) is an aging-dependent, irreversible progressive neurodegenerative disorder and the most common cause of dementia, accounting for an estimated 60–80% of cases worldwide [1,2]. The lack of efficient AD treatments stems from the still incomplete understanding of AD causes and factors modulating AD progression. The prevailing AD hypothesis points to the central role of altered cleavage of the amyloid precursor protein (APP) with an accompanying pathology of the tau protein. While elucidating the role of various factors in the pathological processes associated with these two proteins is still in focus for the identification of novel therapeutic targets, the role of lifestyle factors, including unbalanced nutrition, in the development of AD is now beginning to attract considerable attention.

Over 95% of AD cases are sporadic and late onset, with a complex etiology. Studies indicate that sporadic AD can be triggered by external factors (e.g., environmental and metabolic) and may be associated with other chronic comorbidities, especially those derived from bad lifestyle. Sporadic AD is often accompanied by disorders such as hypercholesterolemia, dyslipidemia, impaired insulin signaling, or impaired cellular energy metabolism [3,4] Furthermore, an increased risk of sporadic AD is associated with the presence of the ε4 allele of apolipoprotein E (APOE ε4) [5], involved in lipid metabolism.

Typical AD brain lesions are senile extracellular plaques composed of Aβ peptides and neurofibrillary tangles (NFTs), which are degenerating neurons filled with over-phosphorylated, aggregated tau protein (p-Tau) [2]. Tau pathology begins in the temporal lobe region of the intraparietal (entorhinal) cortex and gradually progresses, involving structures such as the hippocampus and association cortex. Senile Aβ plaques initially localize in the frontal area of the brain followed by occurrence and deposition in the temporal lobe and later spread into further areas of the brain. Intracellular accumulation of Tau impairs basic neuronal functions essential for survival and leads to a path of degeneration. Tau pathology correlates with progressive gray matter loss and the appearance of the first typical clinical symptoms. In contrast, the initial stage of Aβ plaque formation does not necessarily produce clinical symptoms and is also often observed in patients without symptoms of dementia [6].

### 1.1. p-Tau231 as One of the Earliest AD Biomarkers

Tau belongs to the family of microtubule-associated proteins (MAPs). It is found mainly in the central nervous system (CNS) in glial cells (astrocytes and oligodendrocytes) and in neurons. Tau undergoes numerous post-translational modifications of which the best known is phosphorylation. The balance between tau phosphorylation and dephosphorylation plays a key role in tau function in the cell. Excessive phosphorylation of tau at the threonine 231 residue (Thr231) lowers the ability of tau to bind to microtubules, and may lead to neuronal cytoskeleton disruption; this modification is associated with p-tau pathological aggregation in the cell and is identified in NFTs in AD brains [7,8,9,10,11] In line with this, phosphorylation at Thr231 has been shown to be one of the consistently increased post-translational modifications of tau in the brain that distinguishes neuropathological Braak stages 0–I from III–IV [12,13]. P-Tau(Thr231) is also present in pre-neurofibrillary tangles, prior to overt filament formation, and is described as one of the earliest AD biomarkers in the cerebrospinal fluid (CSF) and plasma of AD patients [10,14]. Plasma p-Tau(Thr231) demonstrates high diagnostic accuracy in identifying subjects at risk for AD and correlates strongly with Aβ-PET and tau-PET, segregating Aβ-PET quartiles better than plasma p-Tau(Thr181) and CSF p-Tau(Thr217) [12]. Furthermore, plasma p-Tau(Thr231) corelates strongly with the tau pathology occurrence in the entorhinal cortex (Braak I–II), one of the earliest regions affected in AD [15].

### 1.2. Western Diet

The term Western diet (WD) describes the consumption of highly processed foods containing simple sugars and saturated fats as the primary components [16,17]. In fact, the “Western diet” defines a broad spectrum of combinations and sources of macronutrients and micronutrients [18,19]. In the exemplary WD, more than half of the supplied energy comes from carbohydrates (51.8%), about 32.8% from fats and 15.4% from protein. In contrast, in the balanced diet, energy provided from protein should account for 19–35%, 22–40% from carbohydrates, and up to 30% from fats. WD consumption is characterized by a high insulin index value and a high glycemic load value [16]. Continuous consumption of high glycemic index foods results in higher average daily blood glucose levels and greater pancreatic insulin secretion [20]. Chronic hyperglycemia and hyperinsulinemia can induce hormonal and physiological changes that promote insulin resistance and the development of other diseases in modern societies [16]. There is ample evidence supporting the utility of appropriate animal models in rodents that mimic the human spectrum of metabolic disorders resulting from WD dietary intake, such as hyperglycemia, hyperinsulinemia, and insulin resistance, among others. Of the models tested, strain C57BL/6 appears to be the most sensitive to WD-induced changes [21,22,23,24,25].

In WD there are also significantly reduced amounts of vitamins and minerals, which promotes avitaminosis and malnutrition of the body despite excessive dietary caloric supply [26,27,28].

### 1.3. Insulin and Insulin Resistance in the Brain

Insulin is the major anabolic peptide hormone produced not only in the β cells of the pancreatic islets of Langerhans, but also by brain cells. The main metabolic stimulus for insulin secretion is an increase in blood glucose levels. The insulin signaling pathway is activated by insulin secretion, which is subject to hormonal, substrate, and neuronal regulation. Insulin is recognized and attached in the body to the insulin receptor (IR) [29,30]. The IR response to insulin is mediated primarily by the insulin receptor substrates (IRS) [31]. Under physiological conditions, insulin attachment to its receptors leads to the activation of the protein kinase B (Akt/PKB) pathway, among others [32]. For example, upon insulin activation, the autophosphorylation of IRS results in the activation of phosphoinositide 3-kinase (PI3K) followed by the downstream activation of phosphorus-dependent protein kinases 1 and 2 (PDK1 and PDK2), which initiates Akt/PKB activation. Upon activation, Akt regulates cell survival, metabolism, and proliferation; affects the organization of the cell cytoskeleton; and leads to the movement of glucose transporters (e.g., GLUT2/4) from the cell interior to the cell membrane, resulting in glucose uptake. Akt is also involved in the inactivation of glycogen synthase kinase-3 (GSK-3). The active beta isoform of GSK3 (GSK-3β) is responsible for the pathological phosphorylation of the tau protein. In turn, normal insulin activity leads to a decrease in GSK-3 activity, which results in less phosphorylation of the tau protein [29,30,31,32,33]. Until recently, it was thought that the brain was not sensitive to insulin. It is now known that binding of the insulin molecule to the IR regulates glucose uptake into the brain through the translocation of glucose transporters. One source of brain insulin is the periphery, where blood insulin enters the brain through the blood–brain barrier (BBB). Insulin transport by capillary endothelial cells in the BBB is affected by many systemic factors, including those related to metabolic disorders, i.e., obesity, systemic inflammation, or diabetes. The second confirmed source of insulin in the brain is its de novo synthesis in brain cells [34,35,36,37]. Insulin in the brain plays many important functions, such as brain–body communication, in which it acts in a counter-regulatory manner to its functions played in the periphery. Brain insulin can participate in the responses to the hypoglycemic state by increasing glucose levels and decreasing serum insulin levels. Insulin in the (CNS) is also a growth factor participating in synaptogenesis and nerve growth, in control of the energy metabolism, neuronal plasticity, neuronal survival under oxygen and glucose deprivation, dynamics of dendrite formation, and increased neuronal activity. Many of these processes play key roles in consolidation of short-term and long-term memory and cognitive function.

Insulin resistance is defined as the lack of sensitivity of target tissues to insulin, even at high insulin concentrations. Insulin resistance is associated with the development of metabolic syndrome including hypercholesterolemia, triglyceridemia, dyslipidemia, obesity, and non-alcoholic fatty liver disease (NAFLD) and results in diseases such as type 2 diabetes (T2D) [38]. In the early stages, the pancreas compensates for the insensitivity of IRs to insulin with increased release of insulin into the bloodstream by pancreatic β-cells. Long-term increased insulin production results in pancreatic cell hypertrophy and hyperinsulinemia [39]. Key players in IR desensitization are substrates for the insulin receptor, specifically IRS-1, whose phosphorylation at serine residue 616 or 307 prevents the insulin molecule from binding to the IR and is a major mechanism leading to insulin resistance [40,41]. Blockade of insulin signaling prevents glucose transport into cells and gradually leads to accumulation of glucose (hyperglycemia) when continuously supplied with food. With feedback, rising blood glucose levels signal an overproduction of insulin in pancreatic cells (hyperinsulinemia), which can result in pancreatic dysfunction and consequently the development of diabetes. The main factor inducing excessive phosphorylation of IRS-1 at serine residues is proinflammatory cytokines. Another factor known to promote this process is the excessive storage of free fatty acids in the obese state [39,40,41,42].

The aim of this study was to verify the hypothesis that WD, by inducing insulin resistance in the brain, can trigger the initiation and propagation of major neuropathological features of AD, such as Aβ plaques and neurofibrillary tangles. To this aim, male C57BL/6J wild-type (WT) mice were fed a WD or standard diet (SD; CTR) from 3 months of age. The effects of WD on brain levels of the insulin signaling pathway components, p-IRS-1(Ser616), p-Akt(473), and p-GSK-3β(Ser9), and on neuropathological AD biomarkers, p-Tau(Thr231) and APP/Aβ levels, were analyzed in animals at 4, 8, 12, and 16 months of age (animal groups 4M, 8M, 12M, and 16M). Levels of these proteins were assessed in two brain structures characteristic of the preclinical, pre-symptomatic stages of AD development—the entorhinal cortex and the hippocampus.

## 2. Results

### 2.1. WD Evokes Hyperinsulinemia, Hyperglycemia, and Systemic Insulin Resistance

Blood glucose levels are dependent on the quantity and quality of carbohydrates consumed in the diet and the functioning of insulin signaling. Elevated blood glucose levels (hyperglycemia) are observed in diabetes and may be indicative of insulin resistance or deficient insulin secretion. Therefore, in this study, blood plasma levels of glucose, insulin, and homeostatic model assessment for the insulin resistance (HOMA-IR) index were determined in all age groups of experimental animals fed a WD and compared to results in mice fed a standard diet (CTR) (Figure 1).

A significant increase in glucose levels in 4-month-old mice fed a WD reveals a hyperglycemic peak. At the same time point, we also observed a significant increase in insulin levels, which translates into an increase in the HOMA-IR ratio indicative of a state leading to peripheral insulin resistance. These observations indicate a strong early response of the body to WD. Figure 1 shows that glucose levels decreased in the older groups to levels comparable to the control groups (4M WD vs. 16M WD), which is likely related to an increase in insulin secretion in response to the WD. In the 8-month-old animals, insulin levels reached a plateau and glucose levels decreased. This indicates a short-term achievement of homeostasis. However, significantly higher insulin levels in the WD group were again observed in the 12- and 16-month-old groups of animals, confirming a steady increase in insulin levels under the influence of the continued WD. The concomitant increase in HOMA-IR in the 12-month-old mice relative to the control group indicates peripheral insulin resistance. It should be noted here that, in the groups from 8M to 16M, the response to the WD varied greatly among individuals, and as a result statistical significance was not obtained.

In summary, in animals maintained on the WD for the shortest period of 3 weeks (4M WD), significant differences between the WD and CTR groups may be related to a strong initial response to a novel adverse stimulus that disrupts homeostasis. After 5 months of WD feeding (8M WD vs. 8M CTR), the stabilization of peripheral insulin and HOMA parameters was observed, probably as a result of strongly interacting compensatory mechanisms and stabilizing homeostasis, despite constant WD stimulus. Subsequently, the results showed a gradual stabilization of the differences between the CTR and WD groups in mice maintained on WD for 9 months (12M WD vs. 12M CTR) and a gradual weakening of protective and compensatory mechanisms with aging, allowing for a worsening of adverse responses to WD-induced changes. The reference ranges of the analyzed parameters were established based on the extreme results for the control group in each age group of mice: for glucose 17.82–23.37 mM/L and for insulin 100.1–147.8 pM/L. Physiologically occurring mean baseline HOMA-IR scores in control mice were 13.22–21.83.

### 2.2. WD Differentially Impacts Insulin Signaling Pathway and Tau Phosphorylation in Entorhinal Cortex and Hippocampus

Next, we analyzed the effects of WD on markers of the insulin signaling pathway and of AD-type neuropathology in the two brain regions: in the entorhinal cortex and the hippocampus. Since AD is age related, we analyzed the influence of both the age factor (4M, 8M, 12M, and 16M) and the diet factor.

The following protein isoforms were analyzed by Western blotting: substrate for insulin receptor 1 phosphorylated at serine 616 site—p-IRS-1(Ser616); protein kinase B/Akt in the active form, i.e., phosphorylated at serine 473 site—p-Akt(Ser473); glycogen synthase kinase in an inactive form, i.e., phosphorylated at the serine 9 site—p-GSK-3β(Ser9); tau protein phosphorylated at the threonine 231 site—p-Tau(Thr231); and full-length unchanged amyloid precursor protein—APP.

In addition, in both brain regions, immunofluorescence analysis of brain tissue sections was performed to assess the presence and location of neuropathological changes typical of AD. The subcellular localization of p-Tau(Thr231) was assessed, because during the initial accumulation of phosphorylated forms of the tau protein, the localization of hyperphosphorylated tau including p-Tau(Thr231) changes. Tau dissociates from microtubules and is progressively displaced from nerve fibers into the cell bodies, leading to destabilization of the cytoskeleton. Analogous brain tissue sections series were immunofluorescently stained to determine whether any changes in APP levels by the WD may be related to amyloidogenic proteolysis of this protein leading to the formation of pathological Aβ peptides in the brain. For this purpose, an antibody, 6E10, specifically recognizing the Aβ peptide was used.

#### 2.2.1. WD Causes Brain Insulin Resistance but Does Not Impact the Level of p-Tau(Thr231) in the Entorhinal Cortex of WT C57BL/6J Mice

Figure 2A,B presents the Western blotting results of mouse brain tissue lysates performed to analyze the insulin signaling pathway in entorhinal cortex of C57BL/6 mice under WD conditions. The data reveal a significant increase in p-IRS-1(Ser616) starting from 8 months of age, which indicates the development of insulin resistance in the entorhinal cortex after approximately 5 months of WD feeding. With respect to peripheral insulin resistance, which was observed in the young, 4M WD group, these data show a delayed response to WD in the brain. When a brief restoration of insulin homeostasis was observed peripherally in 8-month-old mice (Figure 1), data in Figure 2A demonstrate sustained insulin receptor (IR) insensitivity and insulin resistance in the brain at this time.

Analyzing further components of the intracellular insulin signaling pathway, we observed a trend toward decreased levels of the active form of p-Akt(Ser473) in 12-month-old animals fed the WD (Figure 2A,B), which correlates with the increase in the level of p-IRS-1(Ser616) in this group of animals. The pronounced individual variability observed in the levels of p-Akt(Ser473) may be due to the multifunctionality of this kinase and interactions with many other cellular factors.

The next component of the signaling pathway examined is GSK-3β (Figure 2A,B). It was observed that, only in the 12-month-old group, there was a decrease in the level of the inactive form of GSK-3β (p-GSK-3β(Ser9)). This result correlated with a decrease in p-Akt(Ser437) activity and increased level of p-IRS-1(Ser616). In the oldest 16M mice there was a slight rise of the p-GSK-3β(Ser9) level, with a concomitant increase in the level of p-Akt(Ser473) observed, independently of significantly increased level of p-IRS-1(Ser616). The data obtained show that WD causes an insulin signaling pathway downregulation specifically in the entorhinal cortex of 12-month-old mice. However, these WD-induced changes did not affect the level of p-Tau(Thr231) in the entorhinal cortex.

#### 2.2.2. WD Causes Changes in the Location of p-Tau(Thr231) in Neuronal Compartments in Entorhinal Cortex of C57BL/6J Mice

Consistent with the results obtained from Western blotting (Figure 2A,B), the immunofluorescence analysis of sections from the entorhinal cortex area using an anti-p-Tau(Thr231) antibody showed no significant differences under WD (Figure 2C). Although there were no differences in the p-Tau(Thr231) levels, we observed accelerated changes in the localization of the p-Tau(Thr231) isoform in subcellular compartments. The analysis in the youngest groups of animals (4M WD and 4M CTR) revealed a typical pattern of tau protein labeling, both along the course of nerve fibers and in the layer of cytoplasm surrounding the cell nucleus. In 8-month-old mice, such a labeling pattern of p-Tau(Thr231) was observed only in control animals (8M CTR). In contrast, in the 8M WD group, disruption of p-Tau(Thr231) labeling in fibers and restriction of immunostaining mainly to neuronal cell bodies was observed, indicating impairment of normal tau function and dissociation of this protein from microtubules (Figure 2C). In control mice, this phenomenon was age related and appeared in the group of 12-month-old mice (12M CTR) and later in 16-month-old mice (16M CTR).

This result indicates that WD accelerates the alteration of tau protein compartmentalization in neurons and the progressive destabilization of the cell cytoskeleton several months earlier than is observed in the age-dependent progression of tauopathy in control mice.

#### 2.2.3. WD Does Not Cause Insulin Resistance or Changes the Level of p-Tau(Thr231) in the Hippocampus of WT C57BL/6 Mice

In contrast to the results from the entorhinal cortex, in the hippocampal area, we did not observe insulin resistance induction by WD (Figure 3A,B). A significant increase in the level of p-IRS-1(Ser616) was observed only in the 8M WD group of mice in comparison to 4M WD mice. In the older 12M and 16M groups, the p-IRS-1(Ser616) levels decreased in control groups and similarly in WD groups. This indicates that this phenomenon in the hippocampus is dependent predominantly on age but is rather independent of WD. Such observations clearly confirm the different sensitivity of entorhinal cortex and the hippocampus to diet conditions and indicate a specific sensitivity of entorhinal cortex neurons and their quick response to WD-dependent processes resulting in insulin signaling changes.

In the hippocampus only in the 8M WD group, the levels of two assessed kinases, the active form of Akt and the inactive form of GSK-3β, showed a decreasing trend compared with age-matched controls. This is consistent with an increase in p-IRS-1(Ser616) levels in this group. In the other age groups, no differences were observed between animals fed the SD (CTR) and the WD. At the same time, the subtle changes observed in the levels of p-IRS-1(Ser616), p-Akt(Ser473), and p-GSK-3β(Ser9) proteins in the hippocampus were not associated with significant changes in the levels of p-Tau(Thr231) in all assessed age groups. Statistical analysis revealed an aging-related increase only in the level of p-Tau (Thr231) between 8M control mice and 12M control mice.

### 2.3. The Comparison of Entorhinal Cortex and Hippocampus Areas in Terms of WD-Induced Amyloidopathy

The results analyzed so far show that WD specifically induces insulin resistance and impairment of tau protein function typical of progressive tauopathy, selectively in the entorhinal cortex area and not in the hippocampus. In the next step, we compared the effects of WD on both of these brain structures in terms of the induction of the amyloidopathy. To this end, we determined the level of a normal, full-length, amyloid precursor protein (APP) and performed immunofluorescence analysis of the products of pathological amyloidogenic proteolysis of APP—Aβ peptides.

WD Causes a Decrease of Full-Length APP Protein Level Specifically in Entorhinal Cortex but Not in the Hippocampus in WT C57BL/6J Mice

Figure 4A,B shows a WD-induced decrease of the full-length APP protein only in the entorhinal cortex, in the group of deeply aged 16-month-old animals. This WD-induced decrease of APP levels may point to an increasing level of APP degradation as a result of induction of amyloidogenic proteolysis process and the formation of its products in the form of pathological Aβ.

In the hippocampal area we did not observe such significant changes, apart from an apparent decrease in the 4M group fed the WD (Figure 4A,B). In older WD groups, the APP levels reached parity with those observed in control groups. These data again confirm the greater sensitivity of the entorhinal cortex neurons to WD.

Since Western blotting data showed an effect of WD on full-length APP levels in the entorhinal cortex but not in the hippocampus, in the subsequent step, we performed an immunohistochemical analysis of Aβ peptide occurrence only in the sections from the entorhinal cortex (Figure 4C). Regardless of age in all four control groups (4-, 8-, 12-, and 16-month-old groups), the intensity and pattern of Aβ labeling was similar, and only a very weak Aβ signal was observed in the cytoplasm of neurons. In contrast, in 16-month-old mice fed the WD (16M WD), the evaluation demonstrated an intensification of Aβ peptide staining in the cytoplasm of neurons in entorhinal cortex (Figure 4C). This result is consistent with the decrease in APP levels in the 16M WD group of mice shown in Figure 4A. Additionally, in agreement with the results shown in Figure 4A,B, in 4-, 8-, and 12-month-old mice, the WD did not significantly affect either the level of the full-length, normal form of APP, or the change in the labeling pattern of the Aβ peptide in the entorhinal cortex (Figure 4C).

The results presented so far indicate that the first area where the molecular mechanisms leading to the neuropathological changes typical of Alzheimer’s disease are disrupted is the entorhinal cortex. This result is consistent with the existing knowledge on the progression of amyloidopathy and tauopathy processes in the temporal lobe in humans. Additionally, consistent with results from studies in humans, we observed that tau protein-related disorders appeared much earlier than amyloidopathy.

### 2.4. Analysis of WD-Derived Insulin Signaling Impairment on AD Markers, Based on Individual Extreme Values

Next, we performed a detailed analysis of the studied parameters in single individuals. In order to assess the correlation between insulin resistance and tau pathology, we selectively analyzed the individuals with the highest and lowest levels of the p-IRS-1(Ser616) and p-Tau(Thr231). Appendix A presents the extreme values for each individual in all experimental and age groups. Figure 5 shows that some molecular responses to the WD show individual variability and that the differences are the most common and higher in the young 4M animals and stabilize in the older 12M and 16M animals.

#### 2.4.1. Analysis of the Entorhinal Cortex of C57BL/6J Mice Fed with WD

Figure 5A presents analysis of extremes of p-IRS-1(Ser616) and p-Tau(Thr231) in the entorhinal cortex of mice in all age groups. After the initial 3 weeks of WD feeding, 4M animals showed the greatest variation in response to WD and revealed no direct relationship of developing cerebral insulin resistance to levels of p-Tau(Thr231) and APP (Figure 5A, 4M). The individual with the highest level of p-IRS-1(Ser616) showed an increase in p-Akt(Ser473) and p-GSK-3β(Ser9) levels, with no increase in p-Tau(Thr231) and decreased APP levels relative to the mean value for the 4M WD group. In contrast, the highest value for p-Tau(Thr231) correlated with a decrease in APP levels, but these were not associated with impaired insulin signaling in this individual.

The most consistent linear runs and smallest relative min-max ranges for the group were observed in 12-month-old mice after 9 months of WD feeding (Figure 5A: 12M; Appendix A: 12M). In the 12M WD group, there is a stabilization of molecular parameters. In this group, we observed a significant decrease in the level of the active form of Akt, correlating with a decrease in the level of the inactive form of GSK-3β compared to animals on the standard diet (12M CTR, gray bars). However, the disruption of normal activity of both kinases does not translate into an increase in p-Tau(Thr231) and a decrease in APP, either for the range of values for the WD group (blue bars) or for animals maintained on the standard diet (gay bars). The highest level of p-Tau(Thr231) in the entorhinal cortex of 12-month-old mice did not correlate with changes in the levels of any of the other markers tested.

In the entorhinal cortex of the oldest group of 16-month-old C57BL/6J mice (Figure 5A 16M), the increase in the p-IRS-1(Ser616) level correlated with a significant decrease in APP, but did not raise the level of p-Tau(Thr231).

These results confirm that WD influences both p-Tau(Thr231) and APP and may affect the insulin signaling pathway, but the changes are related to differential responses to experimental conditions in particular individuals. It is noteworthy that WD-induced changes in p-IRS-1(Ser616), p-Akt(Ser473), and p-GSK-3β(Ser9) do not necessarily occur simultaneously. A decrease in APP level and likely activation of amyloidogenic proteolysis processes occurs with long-term WD feeding and correlates with an increase of p-IRS-1(Ser616), but this process does not occur simultaneously with tau pathology.

#### 2.4.2. Analysis of the Hippocampus of C57BL/6J Mice Fed with WD

In the hippocampus, the linear patterns of the dependencies of the individual markers assessed are contra-directional compared to those in the entorhinal cortex in all age groups of animals fed the WD (Figure 5B).

In 4-month-old animals (4M WD), the highest level of p-Tau(Thr231) was observed in the individual with the lowest level of p-IRS-1(Ser616). In contrast, a decrease in APP level was observed with the highest level of p-IRS-1(Ser616) for the WD-fed group.

The WD in 12-month-old mice (12M WD) increases the p-IRS-1(Ser616) and decreases p-Akt(Ser473) and p-GSK-3β(Ser9), but neither p-Tau(Thr231) nor APP are significantly affected. In contrast, the highest level of p-Tau(Thr231) was not dependent on changes in the levels of all three insulin signaling markers and was also not associated with a decrease in APP level.

Only in the hippocampus of 16-month-old mice fed with WD (16M WD) was the highest level of p-IRS-1(Ser616) observed consistent with the highest level of p-Tau(Thr231), but unlike the observations from the entorhinal cortex, no reduction in APP levels was observed in these individuals.

**Figure 5 ijms-23-04744-f005:**
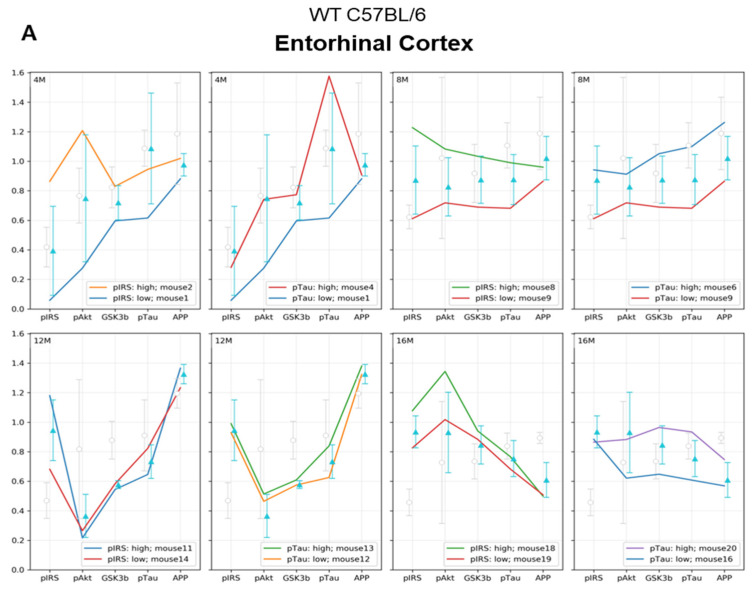
(**A**) Entorhinal cortex and (**B**) hippocampus. Graphs show a linear representation of the relationship of p-Akt(Ser473), p-GSK-3β(Ser9), and APP levels to the minimum and maximum values of p-IRS-1(Ser616) and p-Tau(Thr231). Each colored line represents the results for individuals fed the WD at each age group (4M WD, 8M WD, 12M WD, and 16M WD). The standard deviation ranges for the groups fed the Western diet (WD) are shown in light blue as reference ranges. The standard deviation ranges for the groups fed the standard diet (CTR) are indicated in light gray color as reference ranges. (Analysis of extreme values for each individual in all experimental groups is attached in Appendix A).

From the comparison of the entorhinal cortex and hippocampus in Figure 5, based on the analysis of the extreme values for p-IRS-1(Ser616) and p-Tau(Thr231), it is again apparent that each of these brain structures presents a different sensitivity to the effects of the WD. These detailed analyses of single individuals within age groups consistently showed that the entorhinal cortex is more sensitive to the effects of the dietary WD than the hippocampus. As a result of diet-dependent metabolic abnormalities, the processes of progressive tauopathy and amyloidopathy begin in the entorhinal cortex, probably even before the manifestation of hippocampus-dependent clinical symptoms of AD. Another conclusion from the analysis of extreme values for p-IRS-1(Ser616) and p-Tau(Thr231) confirms that tauopathy and amyloidopathy both may be induced by metabolic disturbances induced by WD through independent cellular processes, not associated with the IRS-1/Akt/GSK-3 pathway.

### 2.5. Comparative Analysis of the Effect of WD on AD Pathology in Wild-Type C57BL/6J Mice and in the AD Mouse Model Tg2576 (APPswe)

Next, we compared the effect of WD on AD pathology in wild-type C57BL/6J mice and in the AD mouse model Tg2576 (APPswe) (Figure 6A,B). In this comparison, we considered animals up to 12 months of age because this age range reflects the early, pre-amyloid stages of AD. Moreover, phenotypic characterization of the Tg2576 mouse model of AD indicates that Aβ plaques appear in these mice at the earliest around 12 months of age. In our previous study, we showed that the deposition of senile plaques in Tg2576 mice begins between 12 and 16 months of age [43]. In the present study, no significant changes in APP levels in the entorhinal cortex were observed in either mouse strain until 12 months of age. As was shown earlier in Figure 4, APP levels decreased only in the entorhinal cortex in 16-month-old C57BL/6J strain mice.

#### 2.5.1. The Levels of p-IRS-1 (Ser616), p-Tau (Thr231), and APP in the Entorhinal Cortex of Wild-Type C57BL/6J Mice and in the AD Mouse Model Tg2576 (APPswe)

As demonstrated in Figure 6A, the analysis of the entorhinal cortex of AD mouse model showed a decrease in p-IRS-1(Ser616) levels with age in the control (CTR) group, similarly as in the WD group. No significant differences in p-IRS-1(Ser616) levels were observed with a weak decreasing trend with age in 12-month-old Tg2576 mice. The exact inverse relationship of p-IRS-1(Ser616) levels with WD intake were observed in wild-type C57BL/6J mice, where we observed a significant increase in p-IRS-1(Ser616) levels in the entorhinal cortex of 8M WD and 12M WD animals compared to age-matched controls and the youngest 4-month-old animals.

The data presented in Figure 6A show that, in the wild-type model, without genome-wide alterations, WD induces insulin resistance, whereas in the AD model, WD has a downregulatory effect on p-IRS-1(Ser616) levels, suggesting a protective role for the transgene, Swedish double mutation (K670N, M671L) in APP gene.

In contrast, when analyzing p-Tau(Thr231) levels, we observed no effect of WD-dependent insulin resistance in wild-type C57BL/6J strain mice, and in the entorhinal cortex of AD Tg2576 mice, there was an increase in p-Tau(Thr231) levels with age, following integration of the transgene in a manner independent of insulin signaling. Only in the youngest group of Tg2576 mice did the analysis show a significant increase in p-Tau(Thr231) under WD compared to control animals, through mechanisms independent of insulin signaling.

#### 2.5.2. The Levels of p-IRS1(Ser616), p-Tau(Thr231), and APP in the Hippocampus of Wild-Type C57BL/6J Mice and in the AD Mouse Model Tg2576 (APPswe)

As presented in Figure 6B, we did not observe any significant dependencies of the levels of studied markers under WD conditions in the hippocampus in both mouse strains. Eventual significant differences are dependent on individual variability and larger or smaller standard deviations shown by the statistical analysis. We revealed only a decreased tendency of p-IRS-1(Ser616) level in 12M Tg2576 mice, similarly as in wild-type 12M C57BL/6J mice. In the hippocampus, we also observed an increase of p-IRS-1(Ser616) in 4M WD group comparing to 8M WD both in WT and Tg2576 mice. Results obtained from p-IRS-1(Ser616) analysis are not related to the results from p-Tau(Thr231) in the hippocampus of either mouse strain. We observed no significant changes in p-Tau(Thr231) level under WD conditions. Full-length APP level in Tg2576 mice significantly increase with age in transgene-dependent manner (Figure 6B). In C57BL/6J mice we didn’t observe such growth in APP levels. Conversely, there is a noticeable decrease during WD feeding as an extreme reaction of C57BL/6J mice to toxic conditions revealed in the youngest 4M mice group. This difference disappears as the WD is applied in these mice (C57BL/6J), suggesting the activation of compensatory processes leading to a return to homeostasis.

### 2.6. WD Influence on p-Tau(Thr231) and Aβ Immunostaining in Entorhinal Cortex of Transgenic Mouse Model of AD

The immunohistochemical analysis of entorhinal cortex tissue section from Tg2576 transgenic mice (Figure 7) revealed a weak intensity of p-Tau(Thr231) staining but typical for its cytoskeletal localization in the cells, in 4M CTR and 4M WD mice. However, in the 4M WD group, a greater accumulation of p-Tau(Thr231) isoform around the cell nucleus was observed. A pattern similar to 4M WD pattern of staining was observed in 8M CTR animals (Figure 7). However, in the 8M WD group, a disappearance of fibers of p-Tau(Thr231) staining was seen, which indicates the start of the disruption of tau functions in microtubule stabilization under WD conditions. In 12M control Tg2576 mice, p-Tau staining only in the cytoplasm of neuronal cell bodies was observed. In 12M Tg2576 mice fed with the WD (12M WD), a significantly higher intensity of cell bodies p-Tau(Thr231) immunostaining was visible when compare to 12M CTR Tg2576 mice. It may additionally indicate a growing cellular production of this p-tau isoform in pathological conditions during WD nourishment.

Immunohistochemical analysis using a 6E10 antibody directed against Aβ peptide (Figure 7) revealed an increasing trend of Aβ deposits around cell nuclei from 4 to 12 months of age in Tg2576 mice maintained on the WD, when compared to control Tg2576 groups. The intensification of Aβ labeling in the transgenic model of AD (Tg2576) is slightly dependent on age, but the WD increases Aβ levels and enhances its cytoplasmic labeling in all age groups of Tg2576 mice tested (4M WD, 8M WD, and 12M WD).

## 3. Discussion

The main research questions posed by this study were whether WD consumption can induce a state of insulin resistance in the brain, and whether this WD-induced insulin resistance can induce molecular mechanisms of AD without any AD-related mutations or AD genetic background in WT animals (C57BL/6J mouse strain).

### 3.1. Insulin Signaling Alteration in AD Human Brain

Disruption of normal insulin signaling and insulin resistance in brain cells (neurons and glial cells) has serious systemic consequences, but is also involved in many neurodegenerative processes, most notably in AD. It has been confirmed that de novo insulin synthesis occurs in the brain and insulin-related disorders in the CNS are not necessarily dependent on peripheral insulin [37,44,45,46]. The highest levels of mRNA expression of the *INS* gene in humans have been confirmed, among others, in the areas of the brain that we study and present in this paper, namely in the temporal lobe region [47,48,49], where the propagation of neuropathological changes typical of AD begins. Reduced insulin expression and signaling mechanisms, expression of insulin mRNA, protein level, insulin receptors, and downstream signaling elements have been demonstrated in patients with sporadic AD. Impaired insulin signal transduction with reduced tyrosine-kinase activity of the IR has also been reported in patients [50,51,52]. Insulin resistance is associated with reduced responses to insulin signaling in the IR/IRS-1/PI3K signaling pathway. In the brain, reduced insulin response is associated with elevations of IRS-1 phosphorylated at serine 616 (p-IRS-1(pS616)) from mild cognitively impaired (MCI) cases to AD cases. The protective role of this hormone against Aβ accumulation is reduced, at the same time as the expression and function of insulin are downregulated by Aβ deposits. The alterations of these effects in AD patients interfere with the neuroprotective actions of insulin, facilitating the brain’s susceptibility to neurodegeneration. Brain insulin resistance is considered an early and common feature of AD, which seems to be closely associated with the IRS-1 dysfunction [31,53].

When considering the effect of WD on impaired insulin signaling in the brain in AD patients, it is important to consider the effect of dietary components such as simple sugars and cholesterol. Studies in AD patients have shown that reduced cerebral glucose and oxygen utilization occurs [54]. Glucose metabolism in neurons in relation to insulin is particularly important for normal brain function. Glucose is critical for brain energy metabolism, and transient or sustained changes in glucose supply can be crucial for neuronal circuit function. Cognitive deficits are associated with insulin resistance, and impairment of insulin-dependent glucose uptake mechanisms can lead to impaired energy metabolism [49]. Using PET technology, reduced glucose metabolism in the temporoparietal cortex, posterior cingulate cortex, and frontal regions in patients has been confirmed [55], which is associated with the occurrence of cognitive impairment typical of AD. Moreover, elevated levels of circulating diet-derived cholesterol increase the risk of AD because cholesterol modulates Aβ synthesis, inhibits the clearance of Aβ, and potentiates the interaction of Aβ with neuronal membranes. Aβ binds to cholesterol to catalyze the formation of oxysterols, which alter the insulin signaling pathway by inhibiting the phosphorylation of the ERK/Akt route [31,56]. Studies on autopsied human frontal cortices have found that the protein content and activity of the brain insulin/PI3K/Akt signaling pathway are significantly decreased in AD patients. Such impairment of the brain insulin/PI3K/Akt pathway leads to the overactivation of GSK-3β, which promotes altered tau hyperphosphorylation and neurodegeneration [57].

The results obtained in this work substantiate observations of the WD’s role in humans. First, we confirmed that the WD indeed resulted in the peripheral metabolic disturbances in WT mice as in the AD mice. Next, we found that the peripheral metabolic disorders in WT mice were followed by induction of an aging-dependent insulin resistance in the brain. In aged WT C57BL/6J mice, the WD led to enhanced amyloidogenic proteolysis of APP, in similar fashion to the APP mutant mice. Despite significant individual variability in wild-type C57BL/6J mice in response to the WD observed in analyses of all parameters examined, these results clearly show that the WD may be considered not only a risk factor, but also a trigger of AD in the absence of AD genetic causes.

In addition, our data revealed the different WD sensitivity of the first brain structures affected in AD, the entorhinal cortex and hippocampus, which are located in the medial part of the temporal lobe of the brain. We found that the entorhinal cortex is more sensitive than the hippocampus to the impairment of brain insulin signaling.

In the entorhinal cortex, WD-induced insulin resistance was followed by progressive tauopathy indicated by changes in p-Tau(Thr231) presence and changes in localization in neuronal subcellular compartments. This result confirms the hypothesis that the PI3K/Akt/GSK-3 signaling pathway underlying insulin resistance in the brain contributes to tau phosphorylation and tauopathy. However, we also observed a WD-induced decrease in APP levels correlating with the appearance of Aβ peptides in the entorhinal cortex of WT animals. Very little data indicate that insulin resistance can increase levels of Aβ peptide in the brain during AD [58]. Our results demonstrated the development of amyloid pathology in parallel to tauopathy in the course of WD feeding in relation to the aging process. The analysis of extreme values for p-IRS-1(Ser616) and p-Tau(Thr231) in our study confirms that tauopathy and amyloidopathy both may be induced by metabolic disturbances induced by the WD through independent cellular processes.

The involvement of particular signaling proteins in the WD-induced brain insulin resistance and the relevance of AD brain biomarkers analyzed in this study for the pathology are discussed below in the light of the existing literature.

### 3.2. The Effect of the Western Diet (WD) on Inducing an Insulin Resistant State in the Brain

In the present study, the areas of the entorhinal cortex and hippocampus from mice brains were analyzed, in order to assess the effect of the WD on the disruption of the insulin signaling pathway. Our results are congruent with the body of reports showing an increase in IRS-1 phosphorylation at the serine (Ser) 312 and 616 sites in AD pathogenesis, whereas under normal signaling conditions, IRS-1 is phosphorylated at the tyrosine site. We observed a WD-derived increase in the level of p-IRS-1(Ser616) in the entorhinal cortex, which indicates the development of insulin resistance. In contrast, the results obtained from our analysis of the hippocampus showed that this structure is insensitive to the WD, where a decrease in p-IRS-1(Ser616) level was observed. There is a report that showed increased levels of p-IRS-1(Ser616) phosphorylation in the hippocampus when C57BL/6J mice were fed with a high-fat diet (HFD) [59]. It is likely that this difference compared to the results of this study is due to the different composition of the diets used. The diet used in the Arnold et al. study did not contain a simple carbohydrate component, and thus the result appears to be due to the effects of only saturated fatty acids. Another study conducted on brain tissue from AD patients showed an increase in serine phosphorylation of IRS-1 in a group with concomitant T2D, indicating an important role of carbohydrate disturbance in these processes [60]. In line with our results, another report showed that 7 days of feeding rats with a WD-type of diet (high-fat-fructose diet, (HFFD)) resulted in a decrease in the level of IRS-1 protein phosphorylation at the tyrosine residue and a decrease in the level of the active form of Akt in the hippocampus [61].

### 3.3. The Effect of the Western Diet (WD) on Kinase Akt Activity

No significant WD-induced changes were observed in the levels of p-Akt(Ser473) in the hippocampus. However, in the entorhinal cortex, a WD-induced decrease in the active form of Akt was observed with aging, in 12-month-old animals. It correlates with an increase in the level of p-IRS-1(Ser616) in the entorhinal cortex in the 12M WD group. A similar correlation was reported in C57BL/6J mice in which a high-fat diet with sugary drinks (HFS) increased the levels of p-IRS-1 protein and caused a decrease in the level of the active form of Akt in lysates from the whole brain [62].

### 3.4. The Effect of the Western Diet (WD) on Kinase GSK-3β Activity

Thr231 is an important priming residue on tau and is a primary phosphorylation site for GSK-3β. The Thr231 residue is located in the assembly domain of tau and phosphorylation of this residue reduced tau affinity for microtubules [10]. Excessive phosphorylation of tau at Thr231 is associated with its pathological aggregation in the cell and is identified in NFTs in the brain of AD patients. Additionally, APP is implicated in regulation of GSK-3 kinase activity. In vitro studies demonstrated that the APP intracellular domain (AICD) interacts with GSK-3 and induces its activity [63]. Analysis of AD human brains showed a reduced level of the inactive form, p-GSK-3, in MCI and late-stage AD (LAD) individuals compared with controls without dementia. Additionally, the phosphorylation of APP at Thr668 was significantly increased in LAD compared to MCI and non-AD individuals, also implying a role for activation of GSK-3 in inducting this phosphorylation. Moreover, the Aβ peptide appearance induces aberrant activation of GSK-3, leading to APP and tau phosphorylation, impaired axonal transport, and microtubule destabilization observed in AD [64]. The present study evaluated the phosphorylation level of GSK-3β, which is directly involved in the hyperphosphorylation of the tau protein in the pathogenesis of AD. There was a decreasing tendency in the level of the inactive form of p-GSK-3β(Ser9) in the entorhinal cortex of 12-month-old animals. These results are consistent with above-described increase in p-IRS-1(Ser616) and decrease of p-Akt(Ser473) levels in this age group of animals.

### 3.5. The Effect of the Western Diet (WD) on Tau Protein Phosphorylation

Increases in GSK-3β kinase activity were shown to result in increased levels of tau protein phosphorylation [62]. Calvo-Ochoa et al. observed increased levels of tau protein phosphorylated by GSK-3β in rats fed a HFFD [61]. Another study using homogenates from the frontal cortex of C57BL/6 mice showed approximately 60% higher levels of p-Tau(Thr231) in the group fed with a HFD diet compared to the control group [65]. In the present study, the quantitative analysis revealed the lack of the effect of WD on p-Tau(Thr231) phosphorylation levels in the entorhinal cortex, despite the confirmed insulin resistance status. Similar observations were made by Grautze et al., who examined the effects of a HFD, high-cholesterol diet (HCD), high-sugar diet (HSD), and a diet containing a combination of all these, on tau protein-related pathology. They conducted the experiment in the entorhinal cortex and hippocampus of hTau mice but did not observe any significant changes in p-Tau [66]. In contrast, in a neuropathological study, colocalization of NFTs and markers of insulin resistance was demonstrated in patients diagnosed with AD [35]. Patients diagnosed with both AD and T2D had more extensive deposition of tau in the new neocortex, as assessed by 11C-PBB3 PET [67]. A lower cerebral metabolic rate of glucose consumption (FDG-PET) has been demonstrated in AD patients and in middle-aged adults at risk for AD, and this parameter is now recognized as one of the neuroimaging methods useful for early AD risk diagnosis [68]. Moreover, peripheral insulin resistance was negatively correlated with gray matter volume in the medial prefrontal cortex, temporal lobe, and parietal areas in AD patients [69]. The divergent reports on this topic may be due to differences in diet composition, differences in the length of feeding, and different responses of individuals to experimental conditions. In addition, wild-type animals like C57BL/6J have high individual variability, which is also evident in results presented in this paper. It is also possible that WD, as a low-grade factor that acts chronically over a long period of time, may produce effects by inducing neuropathological changes that are different across time. Analyzing the proteostatic disequilibrium, pathway disruption, and developed insulin resistance in AD, we can also expect inhibitory activity of protein phosphatase 2A (PP2A) on NFT and Aβ deposition. PP2A contributes to the dephosphorylation of the tau protein, as well as p-IRS at serine residues, and may lead to a return to homeostasis and prevent neurodegenerative processes [33,70]. However, in the present study, the qualitative microscopic imaging analysis revealed a disruption of the compartmentalization of p-Tau(Thr231) inside the neurons in entorhinal cortex under WD conditions. In control animals, age-dependent impairment of p-Tau(Thr231) labeling was observed during aging in 12M and 16M mice. In contrast, in animals fed with WD, the same labeling pattern was observed already in 8M WD. This indicates WD-induced impairment of normal tau protein function and may be associated with progressive breakdown of the cellular cytoskeleton leading to degeneration of neurons in WT animals.

### 3.6. The Effect of the Western Diet (WD) on Amyloidogenic Proteolysis of APP

In WT C57BL/6J mice fed with the WD, we observed a reduction of the level of the full-length form of APP in the entorhinal cortex in the oldest, 16M, WD mice. It may suggest the activation of the process of amyloidogenic proteolysis leading to the formation of the toxic Aβ peptide. Immunofluorescent microscopic analysis revealed an increase in the intensity of Aβ labeling in the cytoplasm in entorhinal cortex neurons also only in the 16M WD group. To compare, no Aβ was observed in any age groups in mice fed with standard chow. This is consistent with the existing knowledge that the mouse, as a species, does not develop Aβ plaques typical for AD in human. The appearance of Aβ labeling in the neuronal cytoplasm under WD in 16M mice indicates that WD can, however, induce the process of amyloidogenic proteolysis in aged WT animals even without genetic determinants.

The data obtained in our study contribute to the current knowledge on the sequence of progression of tauopathy and amyloidopathy in the human brain in the course of AD development. It is known that tauopathy starts in the area of the entorhinal cortex, and later spreads to the hippocampus, which is reflected in the grading of tau pathology propagation according to Braak and Braak [71,72,73]. In the advanced stages of AD, tau pathology spreads to the frontal and parietal lobes. Amyloidopathy starts from the frontal and parietal lobes, and then the pathology spreads to the temporal lobe, occipital lobe, and cerebellum in the most advanced stage of the disease [74,75]. The entorhinal cortex and hippocampus, which are located in the medial part of the temporal lobe of the brain, are the two brain areas primarily responsible for memory functioning, processing, and consolidation of short-term into long-term memory. The entorhinal cortex is directly adjacent to the hippocampus and the neocortex, so it mediates the transfer of information from the neocortex to the hippocampal structure [76,77]. In the entorhinal cortex and hippocampus, neuronal loss occurs at the early stage of AD development [78,79]. While it is known the first brain structures affected in AD are the entorhinal cortex and hippocampus, the comparison of these two structures in terms of the sequence of AD biomarkers’ appearance upon WD or insulin resistance is less known. Our data suggest that WD induces both tauopathy and amyloidopathy first in the entorhinal cortex, and thus changes in this area probably precede the changes in the hippocampus.

Overall, our data support the view that AD can be initiated by metabolic impairment in the periphery [38,43]. It also strongly indicates that a healthy, balanced diet may be one of the most efficient AD prevention methods.

## 4. Materials and Methods

### 4.1. Animal Experimental Groups

Ten-week-old males of C57BL/6J wild-type strain mice were purchased from Medical University of Białystok, Poland and further housed in the Laboratory of Preclinical Testing of Higher Standard at Nencki Institute of Experimental Biology in Warsaw, Poland. Ten-week-old transgenic males of APPswe mice with a Swedish double mutation (K670N, M671L) (Tg line #1349; B6; SJL-Tg (APPSWE) 2576 Kha) were purchased from Taconic Biosciences (Germantown, NY, USA).

Mice were divided into two experimental groups: (1) control (CTR) group fed a standard, balanced diet (SD) starting from 13 weeks of age until they reached 20 months; and (2) mice fed with a Western diet (WD) in the same scheme as the CTR group. Each experimental group was randomly divided into five age groups: 4-, 8-, 12-, 16-, and 20-month-old mice. Due to the increasing mortality with age, completed results are not available for all subgroups. Number of animals included in this article are shown in Table 1 for C57BL/6J mice and Table 2 for APPswe mice.

In order to perform the experiment, the consent of the Local Ethical Committee for Experiments on Animals in Warsaw (LKE) was obtained, and all requirements regarding the treatment of laboratory animals were taken into account and procedures were applied, taking into account the applicable Directive 2010/63/EU of the European Parliament and of the Council of 22 September 2010 on the protection of animals used for scientific purposes and the Polish Act on the protection of animals used for scientific or educational purposes of 15 January 2015 (Journal of Laws of 2015, item 266).

Animals were maintained under a specific pathogen-free (SPF) standard in a system of individually ventilated cages (IVC) (Techniplast, Buguggiate, Italy) and under conditions in accordance with the current Polish Regulation of the Minister of Agriculture and Rural Development of 14 December 2016 on the minimum requirements to be met by the center and the minimum requirements for the care of animals kept at the center.

### 4.2. Diets: Western Diet (WD) and Standard Diet (SD)

The composition of the Western diet (EF R/M E15126-34) and balanced control diet (R/M-H V1534) was from the Ssniff Spezialdiäten GmbH^®^ Company (Soest, Germany) as previously published [43]. The percentage composition of the diets’ ingredients is shown in Table 3.

### 4.3. Mouse Euthanasia and Tissue Collection

Laboratory animals were euthanized by intraperitoneal injection of a mixture of medetomidine at a dose of 1 mg/kg (ORION, Warsaw, Poland; Dexdomitor 0.5 mg/mL) and ketamine at a dose of 75 mg/kg (Biowet, Drwalew, Poland; Ketamine 100 mg/mL) dissolved in 0.9% sterile sodium chloride (NaCl). Blood was collected into heparinized tubes (Sarstedt, Nümbrecht, Germany) from the heart of anesthetized mice for further biochemical analyses, followed by trans-cardial perfusion with cold phosphate-buffered saline (PBS) with 0.1% heparin and 0.184% sodium orthovanadate (Sigma Aldrich, St. Louis, MO, USA). All brain tissue was divided into two hemispheres. Right hemispheres of the brain were used for immunofluorescence staining. Right hemispheres were fixed in 10% buffered formalin, then sections were obtained after embedding in paraffin. From the left hemispheres, the hippocampus and entorhinal cortex were separated to prepare brain tissue lysates for quantifying analysis by Western blot. Tissue was rapidly frozen in liquid nitrogen, then stored at −80 °C before homogenization.

### 4.4. Human Brain Tissue Source

The tissue was obtained from the resources of the Polish Brain Archive of the Institute of Psychiatry and Neurology in Warsaw, Poland. After autopsy, the tissue was stored in 10% buffered formalin until processed in a closed apparatus (Leica HistoCare Pearl Tissue Processor, Leica Microsystems, Wetzlar, Germany). Tissue was rinsed for 24 h in distilled water, then it was dehydrated in a series of increasing alcohol concentrations (6 replicates). The tissue was cleaned in xylene (4 replicates), then saturated with liquid paraffin (62 °C) in triplicate after 2 h and embedded in blocks. The tissue collected in paraffin blocks was cut into sections 8 µm thick using a rotary microtome.

### 4.5. Plasma Total Glucose and Insulin Concentrations

The blood glucose and insulin plasma concentration measurements were collected from mice after 5–6 h of fasting. Plasma was obtained by centrifugation of blood at 4065 rcf for 30 min at 15 °C. Plasma glucose concentration was determined with a biochemical analyzer (Fuji DriChem NX500 Automated Clinical Chemistry Analyzer, Fuji, Tokyo, Japan). Plasma insulin concentration was assayed using Ultrasensitive Mouse Insulin ELISA kit (Crystal Chem, Elk Grove Village, IL, USA; cat. #90080).

### 4.6. Western Blot

Frozen brain tissue was homogenized on ice with a RIPA buffer (Sigma-Aldrich, St. Louis, MO, USA, R0278) along with complete TM Protease Inhibitor Cocktail (Roche, Basel, Switzerland, Cat. No. 04693159001) and a phosphatase inhibitor PhosSTOPTM (Roche, Basel, Switzerland, Cat. No. 04906837001). The obtained homogenate was aspirated into a syringe (20 G 0.9 mm) seven times. Then the test tubes were placed in a rotary mixer (ThermoMixer F1.5, Eppendorf, Hamburg, Germany) at a temperature of 8 °C for 1.5 h. Centrifugation was performed at 15,071 rcf at 4 °C for 15 min (Mini Spin Eppendorf, Hamburg, Germany). The obtained supernatant and pellet were stored at −80 °C.

Protein concentration in tissue lysates was measured using BCA assay (Pierce BCA Protein Assay, Thermoscientific, Waltham, MA, USA). Samples were prepared (20–30 µg per well) using lysates from fragments of brain tissue diluted with water (entorhinal cortex, hippocampus) and using a mixture of Laemmli buffer (BioRad, Hercules, CA, USA) with dithiothreitol (DTT) (Sigma, St. Louis, MO, USA, D9163). Then, electrophoresis was performed using polyacrylamide gels (SDS-PAGE) with 10% tricine or glycine. Proteins from the polyacrylamide gel were transferred to polyvinylidene difluoride (PVDF, Immobilon^®^-PSQ Transfer Membranes, Sigma-Aldrich, St. Louis, MO, USA, ISEQ00005) membrane by semi-dry transfer on a Trans-Blot Turbo Transfer System (BioRad, Hercules, CA, USA). Membranes were blocked for 1.5 h in 5% bovine serum albumin (BSA, Sigma-Aldrich, St. Louis, MO, USA, A3059) in Tris Buffered Saline supplemented with 0.1% Tween (TBST). Membranes were incubated at 4 °C overnight with the following specific primary antibodies: 1:500 pTau Thr231 (GeneTex, Irvine, CA, USA, GTX01563), 1:1000 pIRS1 Ser616 (ThermoFisher Scientific, 44-550G), 1:500 GSK3β (GeneTex, Irvine, CA, USA, GTX86837), 1:1000 pAkt Ser473 (Cell Signaling Technology, Danvers, MA, USA, 4060), 1:1000 anti β-actin (Cell Signaling Technology, Danvers, MA, USA, 8H10D10), and anti-APP C-terminus (Abcam, Cambridge, UK, Y188, ab32136). The horseradish peroxidase-conjugated secondary antibody, 1:5000 anti-rabbit (Cell Signaling Technology, Danvers, MA, USA, 7074) and anti-mouse (Cell Signaling Technology, Danvers, MA, USA, 7076) in bovine serum albumin (BSA, Sigma-Aldrich, St. Louis, MO, USA, A3059) in Tris Buffered Saline supplemented with 0.1% Tween (TBST) buffer, was added to the membrane for a 1 h incubation. Membranes were developed in a ChemiDoc MP Imaging System (BioRad, Hercules, CA, USA) with a Clarity Western ECL Substrate (BioRad, Hercules, CA, USA, 1705061). The intensity of the bands, corresponding to the relative amount of protein, was measured with ImageJ v. 1.46. (Source figures of Western blot analysis attached in Appendix B: Figure A1a,b, Figure A2a,b and Figure A3a,b).

### 4.7. Histological Preparation and Immunofluorescence Staining of Brain Tissue Sections

Collected mouse brain tissues were subjected to a fixation process using 10% buffered formalin using an automated tissue processor (Spin Tissue STP 120, Thermo Scientific, Waltham, MA, USA). The tissue processor subjected the tissue to a series of increasing concentrations (50%, 70%, 80%, and 99.8%) of EtOH, three changes of 99.8% EtOH, three changes of xylene, and two changes to liquid paraffin at 60 °C. Using an embedding workstation (HistoStar, Thermo Scientific, Waltham, MA, USA), paraffin blocks were prepared from the brain fragments and then cut on a rotary microtome (Thermo Scientific, Waltham, MA, USA) to obtain 8 µm thick sections. Tissue was fixed in 10% buffered formalin. Before staining, the brain sections were deparaffinized in three changes of xylene and then rehydrated in decreasing ethanol concentrations (99.8%, 96%, 70%, and 50%) to water. The antigen retrieval was performed in 10 mM sodium citrate buffer (pH 6) with 0.05% Tween in three cycles including a 5 min rinse under elevated microwave temperature set to 600 V, a 10 min cooling at room temperature (RT), and a 5 min cooling on ice. The sections were blocked with 5% normal goat serum with 1% bovine serum albumin in PBS with 0.025% Triton X-100 (PBS-TX) buffer for 1 h at RT. Next, brain slices were incubated overnight at RT with one of the following antibodies: rabbit polyclonal p-Tau (Thr231) 1:100 (Invitrogen, 44-746G) or β-Amyloid (Aβ) (Mouse Purified anti-β-Amyloid, 1-16 Antibody (BioLegend, San Diego, CA, USA, Previously Covance catalog #SIG-39320)). Incubation with a secondary goat anti-rabbit Alexa Fluor 488 and goat anti-mouse Alexa Fluor 594 was performed for 2 h at RT. Cell nuclei were stained with Hoechst. The tissue sections after immunostaining were sealed with Moviol and cover slipped.

### 4.8. Statistical Analysis

Statistical analysis was performed in GraphPad Prism 9 (GraphPad, San Diego, CA, USA). Statistical analysis of the semi-quantitative Western Blot analysis included 3 to 5 patients per CTR or WD group. After checking the normality of the distribution using the Shapiro–Wilk test, a two-way ANOVA with multiple comparisons was performed. A *p*-value < 0.05 was considered a predetermined statistical threshold for significance (* *p* < 0.05, ** *p* < 0.01, *** *p* < 0.001). Completed statistical data are found in Appendix A.

### 4.9. Microphotographs Collection

Microscopic analysis was performed using an automated Nikon Eclipse Ni-E microscope (Nikon, Tokyo, Japan) equipped with a Nikon DSFi1c camera (Nikon, Tokyo, Japan) in NIS-Elements Advanced Research 4.00.00 software (Nikon, Tokyo, Japan). Microphotographs were taken at 400× magnification.

## 5. Conclusions

The entorhinal cortex is more sensitive than the hippocampus to the development of WD-related brain insulin resistance and AD-type pathological changes in the brain of wild-type mice, without any genetic modifications.

WD-induced insulin resistance affects processes involved in amyloidogenic APP proteolysis and Aβ peptide formation in the entorhinal cortex.

The activation of molecular mechanisms leading to the neuropathological changes typical of Alzheimer’s disease—Aβ and NFT—in the entorhinal cortex may precede the changes occurring in the hippocampus.

WD-related metabolic disturbances can induce tauopathy and amyloidopathy through independent cellular processes.

The observed progression of pathological processes leading to NFTs and senile Aβ plaques is age dependent and dependent on the continuity of exposure to a noxious agent.

In summary, these findings highlight the WD as a significant risk factor and a triggering factor for development of insulin resistance and of changes in AD-type pathological markers in the entorhinal cortex, which is believed to be the starting point of Alzheimer’s disease in humans. They point to the possible role of well-balanced diet in the prevention of sporadic AD.

## Figures and Tables

**Figure 1 ijms-23-04744-f001:**
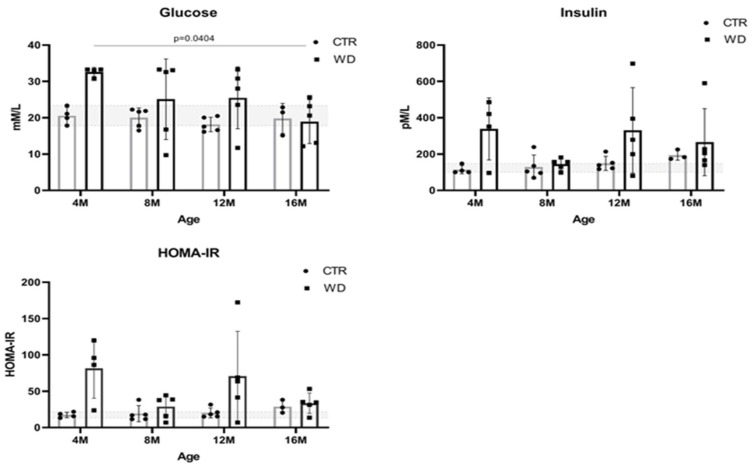
Plasma glucose and insulin levels and HOMA-IR values in C57BL/6J mice fed with a Western diet (WD) and standard diet (SD) in four age groups (4M, 8M, 12M, and 16M). The designated gray boxes show the reference range determined from minimum and maximum values parameters obtained in young (4M) C57BL/6J mice fed standard feed (CTR). Statistical analysis comprised the Shapiro–Wilk normality test, two-way ANOVA, and Tukey’s post hoc test. Significant result: glucose: (4M WD vs. 16M WD (*p*-Value = 0.0404)). (Completed statistical data in Appendix A).

**Figure 2 ijms-23-04744-f002:**
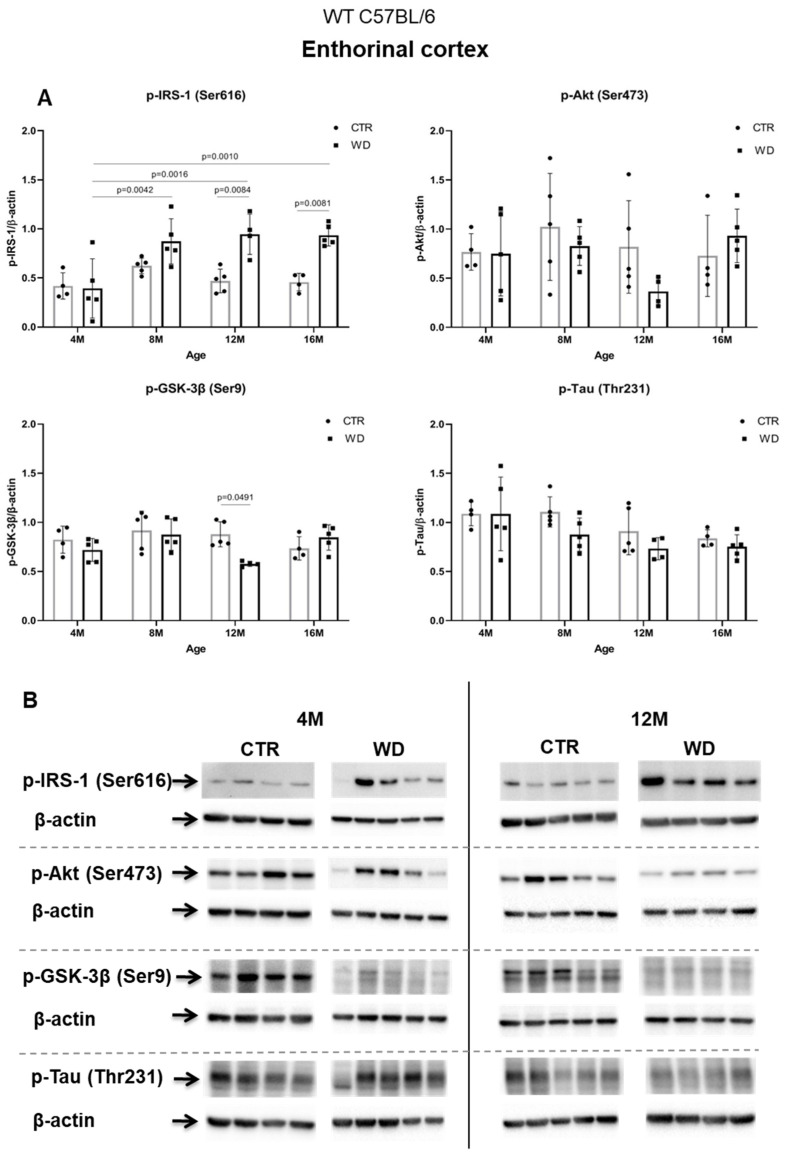
The levels of brain insulin signaling markers and tau phosphorylated at threonine 231 residue in entorhinal cortex of WT C57BL/6J mice. (**A**) Quantification and statistical analysis. (**B**) Representative source Western blots. Statistical analysis comprised the Shapiro–Wilk normality test, two-way ANOVA, and Tukey’s post hoc test; significant results: p-IRS-1 (Ser616): (4M WD vs. 8M WD (*p*-Value = 0.0042); 4M WD vs. 12M WD (*p*-Value = 0.0016); 4M WD vs. 16M WD (*p*-Value = 0.0010); 12M CTR vs. 12M WD (*p*-Value = 0.0084); 16M CTR vs. 16M WD (*p*-Value = 0.0081)); p-GSK-3β (Ser9): (12M CTR vs. 12M WD (*p*-Value = 0.0491)). The source figures from the Western blot analysis are attached in Appendix B: Figure A1a,b; the completed statistical data are in the Appendix A). (**C**) Microphotographs showing representative preparations from the entorhinal cortex of C57BL/6 mouse brains from groups at 4-, 8-, 12-, and 16-months-old, immunofluorescently labeled with an antibody recognizing a tau protein isoform phosphorylated at the threonine 231 site. Negative and positive controls to staining specificity were performed in 20-month-old mice of the transgenic mouse model of AD—Tg2576 (APPswe) and a diagnosed human AD patient. 400× magnification.

**Figure 3 ijms-23-04744-f003:**
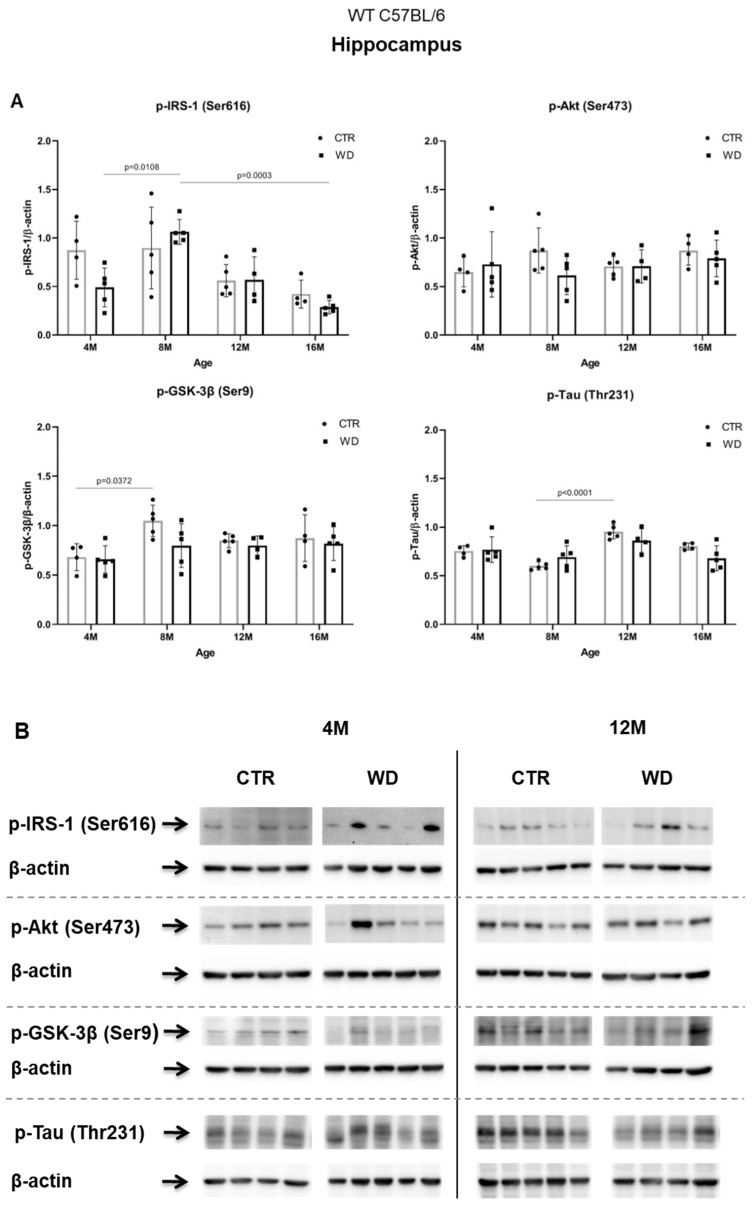
The levels of brain insulin signaling markers and tau phosphorylated at threonine 231 residue in the hippocampus of WT C57BL/6J mice. (**A**) Quantification and statistical analysis. (**B**) Representative source Western blots. Statistical analysis comprised the Shapiro–Wilk normality test, two-way ANOVA, and Tukey’s post hoc test; significant results: p-IRS-1 (Ser616): (4M WD vs. 8M WD (*p*-Value = 0.0108); 8M WD vs. 16M WD (*p*-Value: 0.0003); p-GSK-3β (Ser9): (4M CTR vs. 8M CTR (*p*-Value = 0.0372)); p-Tau (Thr231): (8M CTR vs. 12M CTR (*p*-Value < 0.0001)). The source figures from Western blot analysis are attached in Appendix B: Figure A2a,b; Completed statistical data in Appendix A).

**Figure 4 ijms-23-04744-f004:**
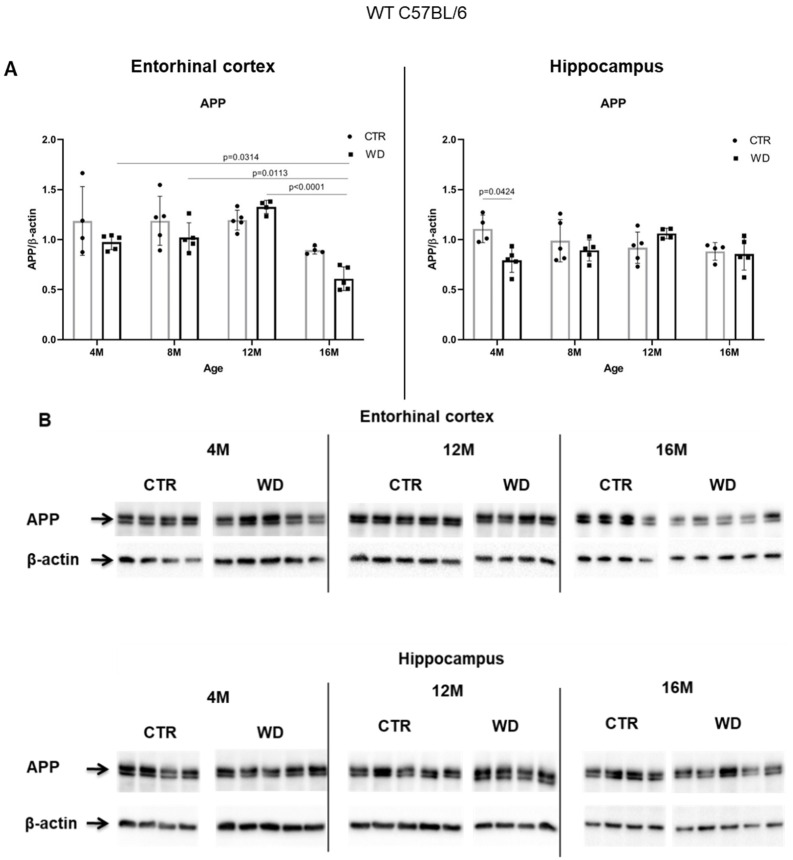
The levels of full-length APP in the entorhinal cortex and in the hippocampus of WT C57BL/6J mice. (**A**) Quantification and statistical analysis. (**B**) Representative source Western blots. Statistical analysis comprised the Shapiro–Wilk normality test, two-way ANOVA, and Tukey’s post hoc test; significant results: Entorhinal cortex APP: (4M WD vs. 16M WD (*p*-Value = 0.0314); 8M WD vs. 16M WD (*p*-Value = 0.0113); 12M WD vs. 16M WD (*p*-Value < 0.0001); Hippocampus APP: (4M CTR vs. 4M WD (*p*-Value = 0.0424)). The source figures from Western blot analysis are attached in Appendix B: Figure A3a,b; completed statistical data are in Appendix A. (**C**) Microphotographs showing preparations from the entorhinal cortex of C57BL/6 mouse brains from groups at 4, 8, 12, and 16 months old, immunofluorescently labeled with an antibody recognizing the Aβ peptide. Negative and positive controls to staining specificity were performed in 20-month-old animals of transgenic mouse model of AD—Tg2576 (APPswe) and a diagnosed human AD patient. Magnification 400×.

**Figure 6 ijms-23-04744-f006:**
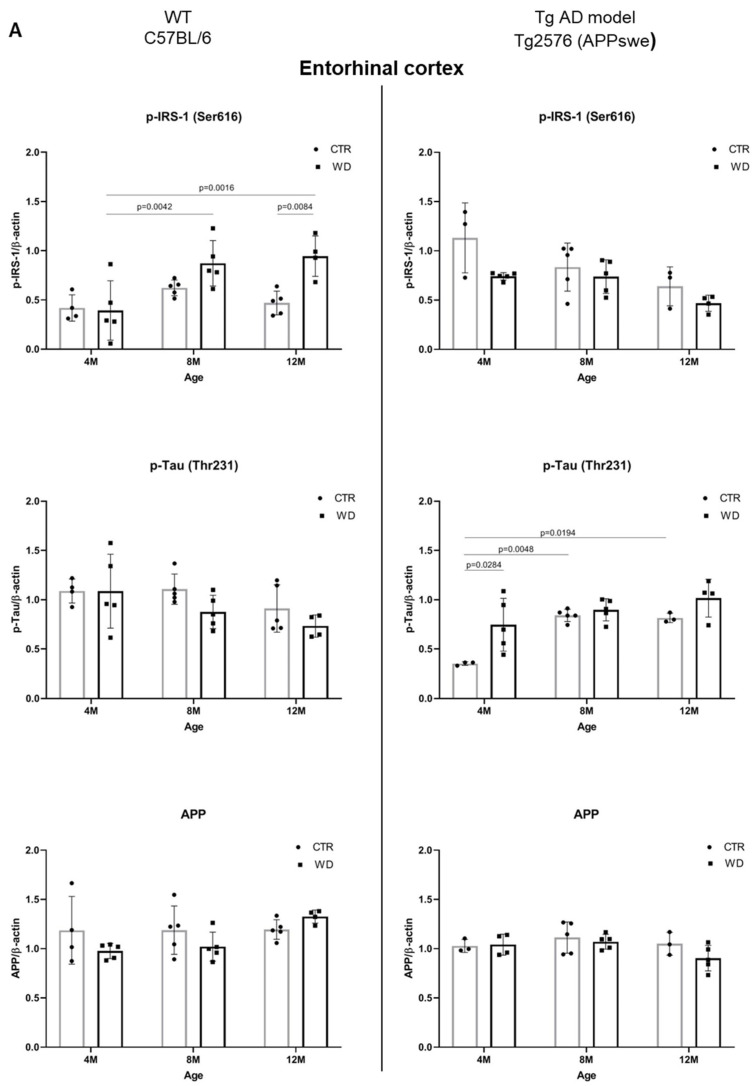
(**A**). Quantitative analysis of the levels of insulin resistance and AD markers in entorhinal cortex—comparison of the AD model Tg2576 (APPswe) and wild-type C57BL/6J. Statistical analysis comprised the Shapiro–Wilk normality test, two-way ANOVA, and Tukey’s post hoc test; significant results: WT C57BL/6 p-IRS-1 (Ser616): (4M WD vs. 8M WD (*p*-Value = 0.0042); 4M WD vs. 12M WD (*p*-Value = 0.0016); 12M CTR vs. 12M WD (*p*-Value = 0.0084); Tg2576 (APPswe) p-Tau (Thr231): (4M CTR vs. 4M WD (*p*-Value = 0.0284)); 4M CTR vs. 8M CTR (*p*-Value = 0.0048); 4M CTR vs. 12M CTR (*p*-Value = 0.0194). Completed statistical data are in Appendix A. (**B**) Quantitative analysis of the levels of insulin resistance and AD markers in the hippocampus—comparison of the Tg AD model Tg2576 (APPswe) and wild-type C57BL/6J. Shapiro–Wilk normality test, two-way ANOVA, and Tukey’s post hoc test; significant results: WT C57BL/6 p-IRS-1 (Ser616): (4M WD vs. 8M WD (*p*-Value = 0.0108)); p-Tau (Thr231): (8M CTR vs. 12M CTR (*p*-Value < 0.0001)); APP (4M CTR vs. 4M WD (*p*-Value = 0.424)); Tg2576 (APPswe) APP: (4M CTR vs. 4M WD (*p*-Value = 0.0277)); 4M CTR vs. 8M CTR (*p*-Value = 0.0059); 4M CTR vs. 12M CTR (*p*-Value = 0.0018). Completed statistical data are in Appendix A.

**Figure 7 ijms-23-04744-f007:**
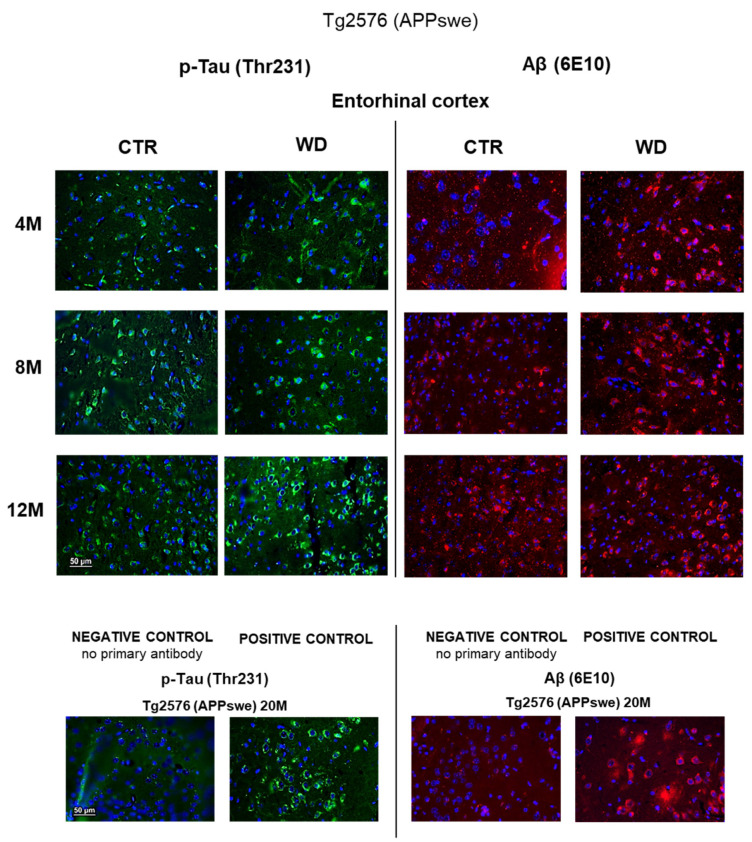
p-Tau(Thr231) and Aβ immunostaining in the entorhinal cortex of the transgenic mouse model of AD in three age groups—4M, 8M, and 12M. Negative and positive controls to staining specificity were performed in 20-month-old transgenic AD model mice—Tg2576 (APPswe). Magnification 400×.

**Table 1 ijms-23-04744-t001:** Number of animals per group in wild-type C57BL/6J mice.

WT C57BL/6	4-Month-Old (4M)	8-Month-Old (8M)	12-Month-Old (12M)	16-Month-Old (16M)
CTR	*n* = 4	*n* = 5	*n* = 5	*n* = 4
WD	*n* = 5	*n* = 5	*n* = 4	*n* = 5

**Table 2 ijms-23-04744-t002:** Number of animals per group in transgenic Tg2576 (APPswe) mice.

Tg2576 (APPswe)	4-Month-Old (4M)	8-Month-Old (8M)	12-Month-Old (12M)
CTR	*n* = 3/4	*n* = 5	*n* = 3
WD	*n* = 5	*n* = 5	*n* = 5

**Table 3 ijms-23-04744-t003:** The composition of the experimental Western diet (WD) and standard diet (SD).

Ingredient	Standard Diet (SD)	Western Diet (WD)
Fat	3.3%	**30.0%**
Fatty Acids	3.25%	**26.87%**
Saturated Fatty Acids	0.57%	**14.19%**
Unsaturated Fatty Acids	2.68%	**12.68%**
Starch	**36.5%**	17.2%
Sugar	4.7%	**16.3%**
Protein	19.0%	20.7%
Cholesterol	-	284 [mg/kg]

## Data Availability

The data presented in this study are available from the corresponding author on request.

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
