# Peer review of "Induction of Brain Insulin Resistance and Alzheimer’s Molecular Changes by Western Diet"

_ijms, 2022, doi:10.3390/ijms23094744_

Round 1

Reviewer 1 Report

The paper of Mietelska-Porowska et al deals with the brain insulin resistance induced by high insulin index and glycemic values of western diet and the effect on molecular pathway involved in Alzheimer’s Disease.

There is a substantial experimental work conducted on animals of different ages. In some points following the presentation of the results is difficult but the discussion helps to better understand the reported data. On the whole, the experimental results support the authors' main conclusions which highlight how diet can constitute a risk factor for Alzheimer’s disease onset and progression.

The paper is of interest to the wide readership of International Journal of Molecular Sciences and deserves publication after minor concerns (see below).

  • The introduction section is too long for an article paper. My suggestion is to reduce it.
  • Report p value also in the Figure 1 caption and not only in the text.

Reviewer 2 Report

Dear authors!

I want to commend the effort going into this experiment and will preface this review with the notion that I have limited experience with mouse models of dementia. As such this review will give the vantage point of a clinician.

Comments:

  1. Introduction: The introduction is well written and sufficiently explains the motivation for this work, but some of the statements need justification.
    1. please provide a reference for line 45-46
    2. please provide background to the comparability of rodent-WD to human WD, e.g. at line 92-93
    3. please provide more background for how insulin resistance or indeed differences in local insulin secretion are altered in the brain in humans

  2. The results section is generally very hard to grasp and confusing to disentangle. My major concern is that, as far as pathological markers of AD are concerned, statistical differences are almost exclusively seen when comparing animals within the same interventional group at different points in time, which is usually assessed by a statistical test that incorporates the dependency, e.g. one-way repeated measures ANOVA. I suggest shortening and restructuring this section to make it more accessible, e.g. grouping the pTau and APP findings, then grouping the histological qualitative assessments, etc.
    The authors continue to perform multiple correlation between essentially all quantified variables, revealing an erratic picture and, mostly, a lack of the expected results. I would again suggest to redact this part and present a concise overview.
  3. Given the concerns, the conclusions drawn are grossly overstated. Indeed, from a pathological perspective, one would expect to see a pattern emerging that resembles the spread along the Braak and Thal stages, which is not observed. Secondly, the lack of phenotypic information limits the applicability to AD as an actual clinical syndrome, since especially in elderly humans, plaques and tangles are readily found without dementia being clinically apparent. I would suggest to rewrite the conclusions to highlight the important aspect that WD influenced only the entorhinal cortex in this experiment, which is believed to be the starting point of disease in humans. From that, speculations might be possible, but I disagree strongly that the data present in this work supports the conclusion that "

    WD as a significant risk factor and a triggering factor for 908 development of sporadic AD. They demonstrate that modifiable lifestyle factors play an 909 important role in the development of neurodegenerative diseases and point to the role of 910 well-balanced diet in the prevention of AD."
